# Defects in Galactose Metabolism and Glycoconjugate Biosynthesis in a UDP-Glucose Pyrophosphorylase-Deficient Cell Line Are Reversed by Adding Galactose to the Growth Medium

**DOI:** 10.3390/ijms21062028

**Published:** 2020-03-16

**Authors:** Christelle Durrant, Jana I. Fuehring, Alexandra Willemetz, Dominique Chrétien, Giusy Sala, Riccardo Ghidoni, Abram Katz, Agnès Rötig, Monica Thelestam, Myriam Ermonval, Stuart E. H. Moore

**Affiliations:** 1INSERM U1149, Université de Paris, 16 rue Henri Huchard, 75018 Paris, France; christelle.arico@gmail.com (C.D.); alexandra.willemetz@inserm.fr (A.W.); 2Institute for Clinical Biochemistry, Hannover Medical School, Carl-Neuberg-Str. 1, 30625 Hannover, Germany; Fuehring.Jana@mh-hannover.de; 3UMR1163, Université Paris Decartes, Sorbonnes Paris Cité, Institut Imagine, 24 Boulevard du Montparnasse, 75015 Paris, France; dominique.chretien@inserm.fr (D.C.); agnes.rotig@inserm.fr (A.R.); 4“Aldo Ravelli” Research Center and Department of Health Sciences, University of Milan, 20146 Milan, Italy; giusy.sala0@alice.it (G.S.); riccardo.ghidoni@unimi.it (R.G.); 5Department of Physiology and Pharmacology, Karolinska Institutet, 17177 Stockholm, Sweden; Abram.Katz@gih.se; 6Department of Cell and Molecular Biology, Karolinska Institutet, 17177 Stockholm, Sweden; monicathelestam@gmail.com; 7Institut Pasteur, Department of Virology, 25 rue du Dr. Roux, 75015 Paris, France; myriam.ermonval@pasteur.fr

**Keywords:** UDP-glucose, UDP-galactose, UGP2, protein glycosylation, prion protein, glycosphingolipid, galactose, congenital disorders of glycosylation (CDGs), developmental epileptic encephalopathies (DEEs)

## Abstract

UDP-glucose (UDP-Glc) is synthesized by *UGP2*-encoded UDP-Glc pyrophosphorylase (UGP) and is required for glycoconjugate biosynthesis and galactose metabolism because it is a uridyl donor for galactose-1-P (Gal1P) uridyltransferase. Chinese hamster lung fibroblasts harboring a hypomrphic UGP(G116D) variant display reduced UDP-Glc levels and cannot grow if galactose is the sole carbon source. Here, these cells were cultivated with glucose in either the absence or presence of galactose in order to investigate glycoconjugate biosynthesis and galactose metabolism. The UGP-deficient cells display < 5% control levels of UDP-Glc/UDP-Gal and > 100-fold reduction of [6-^3^H]galactose incorporation into UDP-[6-^3^H]galactose, as well as multiple deficits in glycoconjugate biosynthesis. Cultivation of these cells in the presence of galactose leads to partial restoration of UDP-Glc levels, galactose metabolism and glycoconjugate biosynthesis. The V_max_ for recombinant human UGP(G116D) with Glc1P is 2000-fold less than that of the wild-type protein, and UGP(G116D) displayed a mildly elevated K_m_ for Glc1P, but no activity of the mutant enzyme towards Gal1P was detectable. To conclude, although the mechanism behind UDP-Glc/Gal production in the UGP-deficient cells remains to be determined, the capacity of this cell line to change its glycosylation status as a function of extracellular galactose makes it a useful, reversible model with which to study different aspects of galactose metabolism and glycoconjugate biosynthesis.

## 1. Introduction

UDP-Glc is required for the biosynthesis of several types of glycoconjugate including glycogen, glycoproteins, glycosaminoglycans and glycosphingolipids (GSL) as well as the polysaccharide hyaluronan (Figure 1). UDP-Glc is generated from glucose-1-phosphate (Glc1P) and UTP by UDP-Glc pyrophosphorylase (UGP). The synthesis of UDP-Gal is catalyzed by enzymes of the Leloir pathway (Figure 1) and requires UDP-Glc, which is either epimerized to UDP-Gal by UDP-galactose 4-epimerase (GALE) or utilized as a uridyl donor by Gal1P uridylyltransferase (GALT). Therefore, UGP is a critical regulator of cellular functions requiring either UDP-Glc or UDP-Gal (Figure 1).

In mammals, UGP is encoded by *UGP2* [1,2], and Chinese hamster (*Cricetulus griseus*, *Cg*) lung (CHL) fibroblasts possessing a point mutation in this gene leading to the defective protein CgUGP(G115D) have been described [3,4,5,6,7]. In the present study, the mutant protein is called CgUGP(G116D) as the previous numbering did not take into account the initiating methionine residue. These cells have been reported to display 4% residual UGP activity, a ~70% reduction in UDP-Glc, and are unable to survive when galactose is presented as the sole energy source [5]. The mutant fibroblasts display altered epithelial-like morphology [7] and present with elevated glucose-regulated protein (GRP) expression [4].

These cells are of considerable interest because their inability to grow when galactose is the only carbon source is also a characteristic of cells from GALT-deficient galactosemia patients in which elevated cellular Gal1P levels are thought to be cytotoxic. Further, after the completion of the work presented here, UGP2-deficient patients presenting with developmental epileptic encephalopathy (DEE) have been described [8]. Therefore, an understanding of the capacity of UGP-deficient cells to metabolize galactose and generate glycoconjugates will be of importance when considering pathophysiological mechanisms and potential treatment strategies for these patients. Here, we report that sugar nucleotide metabolism, galactose metabolism, and glycoconjugate biosynthesis are perturbed in UGP-deficient cells cultivated in normal growth medium. When the mutant cells are transfected with a plasmid encoding wild-type UGP, sugar nucleotide levels, galactose metabolism and glycoconjugate biosynthesis are restored. Unexpectedly, the effects of transfecting UGP-deficient cells with wild-type UGP are to a large extent reproduced by supplementing the growth medium with galactose.

## 2. Results

### 2.1. Rationale for the Study

The dolichol-linked oligosaccharide (DLO) required for protein *N*-glycosylation is glucosylated by enzymes that use dolichyl-P-glucose (DPG) as a donor. The biosynthesis of DPG in turn requires UDP-Glc and led us to ask whether or not conditions that lead to changes in cellular UDP-Glc levels could affect DLO glucosylation and therefore regulate protein *N*-glycosylation in normal cells. Accordingly, characterization of DLO glucosylation and protein *N*-glycosylation in UGP-deficient CHL CgUGP(G116D) fibroblasts possessing 30% normal UDP-glucose levels [5] was of interest. In preliminary experiments, we could not detect DLO glucosylation in these cells (Durrant, C. and Moore, S., unpublished observations), and while attempting to boost UDP-glucose levels by supplementing the culture medium with mixtures of uridine and various monosaccharides, it was noted that supplementation of the culture medium with galactose alone was sufficient to allow complete DLO glucosylation (Durrant, C. and Moore, S., unpublished observations).

This result, coupled with the fact that these cells are unable to survive when galactose is the sole energy source [5], led us to initiate a study of galactose and glycoconjugate metabolism in these cells and provide a global picture of these metabolic pathways in UGP-deficient cells.

### 2.2. Derivation and Description of UGP-Deficient Chinese Hamster Lung Cell Lines

UGP-deficient cell lines were derived as shown in Figure 2A. UGP-deficient Don^Q^ fibroblasts contain reduced levels of mitochondrial cytochrome oxidase (complex IV) when compared to that of parental Don^wt^ cells, but no attempt was made to correlate the mitochondrial deficiency with a reduction in the level of cellular UDP-Glc known to occur in the Don^Q^ cells [9]. In order to investigate a possible link between mitochondrial complex IV deficiency and low UDP-Glc levels, we compared mitochondrial function in Don^wt^, Don^Q^, G3 and Qc cells (see Figure 2A for UDP-Glc and ATP levels in these cells). Data presented in Figure 2B show that Don^Q^, Qc and G3 cells all present with ~70% reduction in complex IV activity and ~50% decrease in complex II activity when compared to those activities measured in Don^wt^ cells. These results demonstrate that the mitochondrial defect in Don^Q^ cells is not correlated with reduced UDP-Glc levels. Further experiments were conducted using G3 and Qc cells, which both have the mitochondrial defect, and for simplicity, they will be refered to as UGP^+^ and UGP^−^ cells, respectively.

### 2.3. Cultivation of UGP^−^ Cells in the Presence of Galactose Causes Changes in Cell Growth and Morphology

Previously obtained data indicate that in normal growth medium supplemented with 5–10 mM glucose and no galactose, UGP^−^ cells, harboring only 4% of parental cell UGP activity, display ~6- and ~3-fold increases in Glc1P and Gal1P levels, respectively [5], compared to parental cells. Substantially larger Glc1P accumulations are probably not observed, as phosphoglucomutase (PGM) assures the interconversion of this metabolite with Glc6P with a K’_eq_ [Glc6P]/[Glc1P] of approximately 20 (Figure 1). The Gal1P increase could be accounted for by inefficient metabolism of galactose. Normal growth media do not contain galactose but serum contains low amounts of this sugar, either free or associated with glycoconjugates susceptible to being internalized and recycled by cells [10]. However, as shown in Figure 1, metabolism of galactose derived from both endocytosed glycoproteins and extracellular galactose requires UDP-Glc. Accordingly, it can be hypothesized that the UGP^−^ cells do not survive when galactose is the only energy source because they cannot generate enough UDP-Glc to metabolize extracellular (cryptic or otherwise) galactose. Data shown in Figure 3A demonstrate that when UGP^+^ cells reach confluence, cell density is a function of the glucose concentration of the growth media employed, but the growth characteristics of these cells do not change when galactose is added to the culture media (Figure 3A). In normal growth medium (11 mM Glc), Qc cells divide more rapidly than UGP^+^ cells during the exponential growth phase. However, after reaching confluence (day 4), while UGP^+^ cells continue to divide, UGP^−^ cell growth rate slows, and after day 6, the numbers of adherent cells decline (Figure 3B, upper panels). Addition of 10 mM galactose to the culture medium allows UGP^−^ cells to reach a higher cell density than that observed in the absence of galactose, but thereafter the number of adherent cells begins to decline (Figure 3B, upper panels). Similar results were obtained with RPMI 1640 medium containing either 1 mM glucose alone or 1 mM glucose and 5 mM galactose. However, under these conditions, as observed for the UGP^+^ cell line, UGP^−^ cells attain lower cell densities than those obtained with media containing 11 mM glucose (Figure 3B, lower panels). While establishing cell growth characteristics as described above, light microscopy observation revealed that galactose has no discernible effects on UGP^+^ cell morphology or monolayer organization (not shown). By contrast, after UGP^−^ cells reach confluence, as shown in Figure 3B, it was noticed that the cells grown in the presence of Gal appear more like fibroblasts and that the monolayer becomes more organized in appearance. This is not simply a consequence of increased cell density because UGP^−^ cells grown in 11 mM glucose alone reach a similar cell density to those grown in 1 mM glucose and 5 mM galactose, but do not display the above-described morphological changes, which therefore appear to be galactose-induced. Because changes in UGP^−^ cell morphology are predominant at confluency, a period when glycoconjugate-mediated cell/cell contacts are likely to be important, glycoconjugate biosynthesis in the two cell lines cultivated in either the presence or absence of galactose was investigated.

### 2.4. Defective Galactosylation of O-, and N-Glycans in UGP^−^ Cells

UDP-Gal is an important sugar donor involved in the Golgi apparatus-situated maturation steps of both *O*- and *N*-linked glycans. The *Ricin communis* agglutinin I (RCA-I) lectin reacts with terminal non-reducing galactose residues of both *N*- and *O*-glycans and lectinoblot studies using this lectin revealed little staining in UGP^−^ cells compared to UGP^+^ cells (Figure 4A). RCA-I reactivity was substantially restored in galactose-cultivated UGP^−^ cells (Figure 4A). The jacalin lectin [11] was then employed to specifically evaluate *O*-glycosylation in these cell lines. Although this lectin is specific for *O*-glycans, it can react strongly with truncated *O*-glycan structures such as the Tn antigen (see Figure 4B for summary of *O*-glycan structures mentioned here) or the core 3 structure, which require neither UDP-Glc nor UDP-Gal for their biosynthesis [12]. In fact, proteins derived from UGP^−^ cells revealed a different pattern of jacalin reactivity (both in terms of intensity and distribution of the bands) than that seen in the UGP^+^ cells (Figure 4C). These differences in protein *O*-glycosylation are partially reversed when UGP^−^ cells are cultivated with galactose. In order to positively identify truncated *O*-glycans in UGP^−^ cells, Western blot analysis was performed using a monoclonal antibody specific for the Tn antigen (Figure 4B) [13]. As expected from studies of other cell lines, Figure 4D shows that UGP^+^ cells display little anti Tn reactivity. By contrast, UGP^−^ cells reveal strikingly raised levels of this antigen associated with glycoproteins with a molecular mass range between 25 and 177 kDa, with particularly strong reactivity of glycoproteins of approximately 36 and 80 kDa (Figure 4D). Cultivation of UGP^−^ cells with galactose almost reduced Tn reactivity to that seen in UGP^+^ cells (Figure 4D). In order to evaluate galactosylation of *N*-glycans, both cell lines were radiolabeled with [2-^3^H]mannose for 18 h in 0.5 mM Glc and [2-^3^H]mannose-labeled glycopeptides were prepared from both the cells and radiolabeling media. Glycopeptides derived from UGP^−^ cells and medium were first submitted to RCA-I-agarose chromatography and shown to yield 4–6-fold less *N*-glycans bearing terminal non-reducing galactose residues than those derived from UGP^+^ cells (Appendix A). When UGP^−^ cells were metabolically radiolabeled in medium containing 0.5 mM Glc and 1.0 mM Gal, the patterns of *N*-glycan galactosylation were substantially normalized (Appendix A). Analysis of the same glycopeptide fractions by ConA-Sepahrose chromatography revealed a significantly higher proportion of polymannose-type glycopeptides in UGP^−^ cells compared to UGP^+^ cells. Interestingly, although galactose treatment of the UGP^−^ cells reduced the proportion of polymannose-type glycopeptides to the level seen in UGP^+^ cells, the proportion of complex-type glycopeptides was not restored to that seen in UGP^+^ cells, and instead there was a corresponding increase in the proportion of hybrid-type glycopeptides (Appendix A).

### 2.5. Impaired Glycosphingolipid Biosynthesis Is Reversed When UGP^−^ Cells Are Cultivated in the Presence of Galactose

UDP-Glc and UDP-Gal are also important sugar donors for the biosynthesis of glycosphingolipids, and a reduction in cellular GM3 (NeuAcα2,3Galβ1,4Glcβ-ceramide) occurs in UGP^−^ cells [14]. Glycosphingolipid biosynthesis begins by the synthesis of glucosylceramide from ceramide and UDP-Glc by glucosylceramide synthase [15]. Figure 4E shows that, unless galactose is added to the culture medium, glucosylceramide is apparent in UGP^+^ cells but not in UGP^−^ cells. Compared to that observed in untreated UGP^+^ cells, the glucosylceramide level also increased in galactose-treated UGP^+^ cells. Although the formation of lactosylceramide (Galβ1,4Glcβ-ceramide) from glucosylceramide requires UDP-Gal, differences in the level of this compound could not be detected in either UGP^+^ or UGP^−^ cells irrespective of the presence of galactose in the growth medium (Figure 4E). By contrast, GM3, the predominant ganglioside expressed by fibroblasts [15], was not detected in UGP^−^ cells unless galactose was added to the growth medium (Figure 4E,F).

### 2.6. UGP^−^ Cells Are Unable to Glucosylate Dolichol-Linked Oligosaccharide Required for Protein N-Glycosylation

The dolichol-linked oligosaccharide (Glc_3_Man_9_GlcNAc_2_-PP-dolichol: DLO) requires the addition of three residues of glucose in order to become an efficient substrate for oligosaccharyltransferase (OST)-mediated protein *N*-glycosylation. These glucose residues are added by dolichol-P-glucose (DPG)-requiring glucosyltransferases. DPG itself is synthesized by the UDP-Glc-requiring DPG synthase. As shown in Figure 5A, glucosylated DLO forms (G_1–3_) are not detected in UGP^−^ cells unless galactose is added to the culture medium.

In patients with congenital disorder of glycosylation (CDG) associated with mutations in the *ALG6* gene, which encodes Man_9_Glc*N*Ac_2_-PP-dolichol: dolichol-Glc glucosyltransferase, Man_9_Glc*N*Ac_2_-PP-dolichol is poorly glucosylated and inefficient protein *N*-glycosylation ensues [16]. Accordingly, we looked at the efficiency of prion protein (PrP^C^) *N*-glycosylation in UGP^−^ cells. The GPI-linked prion protein is normally synthesized as several glycoforms containing 0, 1 or 2 *N*-glycans, and, as confirmed by the polydisperse appearance of the migration profile of PrP^C^ in UGP^+^ cells (Figure 5B,C), each site is occupied by a complex array of *N*-glycans [17]. In UGP^+^ cells, the predominant PrP^C^ glycoforms bear 2 *N*-glycans, and are found on the cell surface (Figure 5D) and in the culture medium (Figure 5B) after release from the plasma membrane by phospholipase C. In UGP^−^ cells, the major PrP^C^ species is not *N*-glycosylated and bands corresponding to glycoforms migrating as mono *N*-glycosylated PrP^C^ are apparent in the cell extracts (Figure 5B,C). The recovery of hypoglycosylated PrP^C^ from the UGP^−^ cell surface and media compartments is reduced compared to that of fully glycosylated PrP^C^ recovered from the same UGP^+^ cell compartments. This is supported by surface immunofluorescence staining of PrP^C^, which is substantially reduced in UGP^−^ cells when compared to that seen in UGP^+^ cells (Figure 5E). Cultivating UGP^−^ cells in normal growth medium containing 10 mM galactose (Figure 5C,D) largely corrects the abnormal PrP^C^ glycosylation profile. Cell surface biotinylation (Figure 5D) and cell surface immunofluorescence (Figure 5E) experiments indicate that as well as normalizing PrP^C^ glycosylation, cultivation of UGP^−^ cells with galactose allows the correctly glycosylated prion protein glycoform to be transported to the cell surface.

### 2.7. [6-^3^H] Galactose Metabolism Is More Efficient in Galactose-Treated UGP^−^ Cells than in Their Non-Treated Counterparts

The above-described data demonstrate that all the defective glycoconjugate biosynthetic pathways investigated can be partially restored by cultivating UGP^−^ cells in the presence of galactose and suggests a common underlying mechanism for this unexpected phenomenon. In order to characterize this mechanism, we investigated the capacity of UGP^−^ cells to metabolize extracellular galactose. UGP^+^ and UGP^−^ cells were cultivated in media containing either 1 mM Glc or 1 mM Glc + 5 mM Gal for 12 h prior to metabolic radiolabeling with [6-^3^H]galactose for 30 min. Cell extracts were subjected to solid-phase extraction (SPE). Radioactivity associated with the SPE flow through material, known to contain monosaccharides and monosaccharide phosphates [18], was further fractionated by diethylaminoethyl (DEAE) anion-exchange chromatography. Charged components were eluted from the column, and, subsequent to alkaline phosphatase digestion, were resolved by TLC. The only radioactive component detected was galactose, indicating that the original structure possessed characteristics expected of [^3^H]galactose-1-phosphate ([^3^H]Gal1P). Radioactive sugar nucleotides were eluted from the SPE columns as described in Materials and Methods and analyzed by HPLC. UGP^+^ cells generated UDP-[^3^H]Gal and UDP-[^3^H]Glc without detectable amounts of [^3^H]Gal1P (Figure 6A). By contrast, UGP^−^ cells generated >100-fold less UDP-[^3^H]Gal and UDP-[^3^H]Glc, and an accumulation of [^3^H]Gal1P was observed (Figure 6B). Pretreatment of UGP^−^ cells with 1 mM Glc + 5 mM Gal for 12 h prior to metabolic radiolabeling caused a substantial reduction in radioactivity associated with [^3^H]Gal1P and a 10-fold increase in UDP-[^3^H]Gal and UDP-[^3^H]Glc (Figure 6D). Pretreatment of UGP^+^ cells with 1 mM Glc + 5 mM Gal for 12 h did not greatly change the amount of UDP-[^3^H]Glc generated, but did provoke a 1.4-fold increase in UDP-[^3^H]Gal (Figure 6C). Therefore UGP^−^ cells can metabolize [6-^3^H]galactose to form UDP-[^3^H]Gal and UDP-[^3^H]Glc, and this metabolism becomes more efficient after ‘conditioning’ the cells with galactose.

These metabolic pulse radiolabeling experiments suggest that the steady state level of both UDP-Glc and UDP-Gal should increase upon cultivation of UGP^−^ cells with galactose.

### 2.8. UGP^−^-Deficient Cells Contain Less than 5% Normal UDP-Glc Levels and Extracellular Galactose Partially Restores UDP-Glc in These Cells

Nucleotide sugar levels were assayed in UGP^−^ and UGP^+^ cells after cultivation under our standard conditions. Whereas UDP-Glc and UDP-Gal occurred in the normal range in UGP^+^ cells, greater than 20- and 10-fold reductions, respectively, in the levels of these compounds were observed in UGP^−^ cells (Figure 6E). Despite the strikingly reduced UDP-Glc/UDP-Gal levels, UGP^−^ cells routinely displayed 1.5–2-fold increases in UDP-Gal*N*Ac and UDP-Glc*N*Ac (UDP-Hex*N*Ac), and 8–10-fold increases in an unknown compound (marked with asterisk). Incubation of UGP^−^ cells with 5 mM galactose in the presence of 1 mM glucose enabled partial restoration of both UDP-Gal and UDP-Glc levels (Figure 6F). Furthermore, under these conditions the levels of UDP-Hex*N*Ac and the unknown compound marked with an asterisk were normalized.

### 2.9. The Ability of Galactose to Increase Cellular UDP-Glc in the UGP^−^ Cells Is Inhibited by Glucose, Does Not Require Protein Synthesis and Is Rapidly Reversible

Data shown in Figure 7A show that, in the absence of exogenous galactose, the level of UDP-Glc in UGP^−^ cells is insensitive to changes in extracellular glucose (0–10 mM) concentrations. By contrast, incubating UGP^−^ cells with 5–10 mM galactose in the absence of exogenously added glucose normalizes cellular UDP-Glc levels with respect to UGP^+^ cells (Figure 7A). The effect of galactose on UGP^−^ cell UDP-Glc levels is blocked by the simultaneous presence of glucose in the incubation medium (Figure 7A).

However, as shown in Figure 7B, the effect of galactose in UGP^+^ cells was rapid. However, in UGP^−^ cells, the response was slower, with a 1 h lag period before detection of an increase in UDP-Glc. In order to understand the origin of this lag period, we asked whether the effect of galactose required protein synthesis. To this end, cells were pretreated with the protein synthesis inhibitor cycloheximide (CHX) 1 h before initiating the galactose treatment. As shown in Figure 7B, CHX did not alter the ability of galactose to augment UDP-Gal and UDP-Glc levels in the UGP^−^ cells. Next, the reversibility of the galactose-induced increase in cellular UDP-Glc levels was investigated in UGP^−^ cells, and as shown in Figure 7C (open symbols), the UDP-Glc level falls rapidly after the initiation of a galactose-free chase. DLO turnover is rapid and if DLO hypoglucosylation in UGP^−^ cells is solely due to lack of UDP-Glc the glucosylation of DLO should reflect changes in UDP-Glc levels. As shown in Figure 7C (closed symbols) and D, the onset of galactose-promoted DLO glucosylation follows closely the galactose-induced increase in UDP-Glc. Similarly, when galactose is removed from the incubation medium, DLO glucosylation and the UDP-Glc concentration fall similarly. Cell surface glycoconjugates turnover more slowly than DLO, and the RCA lectinoblots shown in Figure 7E indicate a lag period of at least 4h before the galactose-induced increase in RCA binding to glycoproteins can be detected in UGP^−^ cells: with the maximal effect taking longer than 19 h.

### 2.10. Determination of the Kinetic Constants of the HsUGP(G116D) Variant with either Glc1P or Gal1P as Substrate

The lag period required for exogenous galactose to increase cellular UDP-Glc/Gal may be explained by the time-dependent accumulation of Gal1P when UGP^−^ cells are treated with galactose such that after 1–2 h, there might be a high enough Gal1P concentration to allow potential UGP-mediated UDP-Gal generation. It has previously been demonstrated that in fibroblasts derived from GALT-deficient galactosemia patients [19] and GALT-deficient yeast [20], the cytostatic effect of extracellular galactose can be reduced by overexpressing UGP [21]. Although human UGP was reported to have a K_m_ of ~20 μM for Glc1P and a K_m_ of ~2 mM for Gal1P, the V_max_ for these substrates appeared to be similar, indicating that high levels of Gal1P could lead to significant UGP-mediated UDP-Gal generation [21]. Although UGP^−^ cells display a ~20-fold reduction in UGP activity compared to that observed in parental Don^wt^ cells, it should be noted that in other systems residual enzyme activities of less than 5% are capable of generating near normal levels of product [22,23]. It is also possible that the altered kinetic constants of the UGP(G116D) variant toward its substrates favor the metabolism of Gal1P. In order to answer these questions, enzyme kinetic studies were performed on purified recombinant human wild-type UGP (*Hs*UGP) and *Hs*Ugp(G116D). Of the 508 amino acid residues in *Hs*UGP and *Cg*UGP, 498 residues are identical (98% identity), and of the 10 non-identical residues—none of which are situated near the active site—6 correspond to conservative substitutions. As shown in Figure 8A, the small increase in the K_m_ of HsUGP(G116D) for UTP compared to the wild-type enzyme is within the range of error. On the other hand, the K_m_ of HsUGP(G116D) for Glc1P is significantly increased, but, being in the double-digit micromolar range, is still low. While substrate affinities are only modestly affected, the mutation has a major effect on V_max_, which is decreased by a factor of approximately 2000 (Figure 8A,B). Figure 8C shows the substrate–velocity curves of the wild-type HsUGP with Gal1P as a substrate. The V_max_ of wild-type HsUGP with Gal1P/UTP is reduced by a factor of approximately 200 when compared to its V_max_ with Glc1P/UTP. As the curve differs slightly from the typical Michaelis–Menten model, the data are also fitted using a cooperative model. The curve fit gives a Hill coefficient of approximately 1.3, which could indicate very mild positive cooperativity for Gal1P. Wild-type HsUGP, an octameric enzyme, also displays a similar mild cooperativity for its substrate, PP_i_, in the reverse reaction [24]. However, under the assay conditions employed, we were unable to detect production of UDP-Gal from Gal1P by HsUGP(G116D).

## 3. Discussion

### 3.1. UDP-Glc Levels in UGP^−^ Cells

Previous studies [3,4,5] have shown that Don^Q^ and Qc cells possess 25–35% of the quantity of UDP-Glc found in Don^wt^ and G3 cells (see Figure 2A for derivation of cell lines). However, under our normal culture conditions, UGP^−^ (Qc) cells present with less than 5% the UDP-Glc level found in UGP^+^ (G3) cells. The origin of these differences is not clear but may well reflect the different culture or assay conditions used in the different studies. In fact, in control and galactose-treated UGP^+^ cells and galactose-treated UGP^−^ cells, UDP-Glc/UDP-Gal and UDP-Hex*N*Ac levels fall 3–4-fold during the cell growth cycle (Moore, S., unpublished data). Furthermore, as we show that UDP-Glc/UDP-Gal levels in UGP^−^ cells depend upon the ratio of glucose and galactose concentrations in the culture medium, the basal levels of these nucleotide sugars may be affected by the amounts of endogenous sources of galactose in the fetal calf serum used [10], especially after cells have depleted the glucose of the growth medium. We noted a 1.5–2-fold increase in UDP-Glc*N*Ac/UDP-Gal*N*Ac and an 8-fold increase in a component (indicated with an asterisk in Figure 6 and Figure 7) whose elution position was similar to, but not the same as, CMP-Sia in UGP^−^ cells. Treatment of UGP^−^ cells with galactose normalized the levels of these components with respect to their levels in UGP^+^ cells cultivated under standard conditions. However, it was noted that treating UGP^+^ cells with galactose also caused a 50% reduction in the level of UDP-Hex*N*Ac compared to the level seen in these cells cultivated under standard conditions. The identity of the component with a similar retention time to CMP-Sia is being investigated. The CMP-Sia standard used in Figure 7 is CMP-*N*-acetylneuraminic acid (CMP-Neu5Ac), which is the major CMP-Sia found in Chinese hamster ovary (CHO) cells. Much smaller amounts of CMP-*N*-glycolylneuraminic acid (CMP-Neu5Gc) have also been reported to occur in CHO cells [25].

### 3.2. Defective Galactose Metabolism

When tissue culture medium containing glucose is supplemented with galactose, striking increases in cellular UDP-Gal levels are seen in UGP^+^ cells (Figure 7B) and these results are similar to those reported for normal skin biopsy fibroblasts cultivated in the presence of glucose and galactose [26]. This effect is thought to underlie the normalization of glycosylation profiles seen in patients with various glycosylation defects that have been treated with galactose [26,27,28]. Here, we demonstrate that UGP-deficient cells can also metabolize galactose: first, UGP^−^ cells can, albeit inefficiently, generate UDP-[6-^3^H]Gal/UDP-[6-^3^H]Glc when pulse radiolabeled with [6-^3^H]galactose; second, UGP^−^ cells metabolize a pulse of [6-^3^H]galactose more efficiently after the cells have been pretreated with galactose; third, steady state levels of UDP-Glc/UDP-Gal in UGP^−^ cells are increased after prolonged cultivation in the presence of galactose. How can these observations be explained? Metabolism of galactose via enzymes of the Leloir pathway requires UDP-Glc. However, other mechanisms for galactose metabolism not requiring UDP-Glc have been proposed. There have been suggestions of a UDP-galactose pyrophosphorylase activity independent of UGP [29]. However, classical biochemical studies indicate that cellular UDP-Glc and UDP-Gal pyrophosphorylase activities reside in the same protein [30]. After some uncertainty (see http://www.uniprot.org/uniprot/Q16851), it now appears that mammalian genomes harbor a single gene (*UGP2*) encoding UGP [2]. Presently therefore, the presence of a *UGP2*-independent UDP-galactose pyrophosphorylase activity in UGP^−^ seems unlikely. Although some organisms harbor a UDP-sugar pyrophosphorylase that can generate UDP-Glc and UDP-Gal from Glc1P and Gal1P, respectively, only plants and protists possess such an activity [31]. So, either the 5% residual UGP activity reported to occur in UGP^−^ cells allows sufficient galactose metabolism, or treating cells with galactose somehow facilitates galactose metabolism by altering enzyme activities involved in the control of UDP-Glc/UDP-Gal. With the former hypothesis in mind, we assayed UGP activity in extracts derived from Don^wt^ cell (see Figure 2A for derivatization of cell lines) extracts and UGP^−^ cells cultivated for 6 h in media containing either 1 mM glucose or 1 mM glucose + 5 mM galactose. Although we noted similar UGP activities in galactose-treated Don^wt^ cells and their non-treated counterparts, UGP activity was below the limit of detection in either the control or galactose-treated cell extracts using our assay procedure (Moore S, unpublished data). The very low UGP activity in UGP^−^ cell extracts is in accordance with kinetic data obtained in vitro using the recombinant HsUGP proteins (Figure 8). It is demonstrated that the V_max_ of the G116D variant is 2000-fold less than that of the wild-type protein with the natural substrates Glc1P and UTP. Furthermore, the V_max_ of the wild-type protein with Gal1P/UTP as substrates is 200-fold less than that with Glc1P/UTP. Finally, the activity of the HsUGP(G116D) variant with the Gal1P/UTP substrates was too low to be measured. Because we wanted to understand how the G116D mutation affects affinities for the natural substrates Glc1P and UTP, we determined the respective K_m_ values. The HsUGP(G116D) variant has a significantly increased (but still low) K_m_ for Glc1P compared to that of the wild-type enzyme, while the K_m_ for UTP is not significantly affected. The Km values obtained by Lai et al. are different from those reported here but the tendency and order of magnitude is comparable [21]. However, our V_max_ values are much higher than those reported in Lai et al. and, most importantly, we see a greatly reduced V_max_ with Gal1P compared to Glc1P. These differences may arise from either the different protein preparations used or from the different assays employed. In the previous study, end-point determinations employing different coupled reactions to detect UDP-Glc or UDP-Gal were used [21], whereas we used a continuous assay based on PPi detection, which is independent of Glc/Gal. Although G116 is located within the nucleotide-binding region of the active site [24], its mutation may affect Glc1P binding indirectly. Since the enzyme follows an ordered sequential Bi-Bi mechanism with UTP as the first substrate, it can be speculated that HsUGP(G116D) binds UTP with nearly wild-type affinity, but its altered orientation within the active site may prevent efficient subsequent binding of Glc1P and in consequence dramatically affect catalysis. The physiological consequences of the small increase in the K_m_ of the mutant enzyme for Glc1P is not obvious, but it should be noted that under normal culture conditions, the Glc1P level is approximately 6-fold higher in UGP-deficient cells compared to that in the wild-type cells: suggesting that there may be sufficient binding of Glc1P to UGP(G116D). To conclude, from this data, it is difficult to understand how the extremely low residual UGP activity could account for UDP-Glc/Gal production in galactose-treated UGP^−^ cells. The delay in the onset of the galactose rescue phenomenon maybe related to the accumulation of Gal1P in galactose-treated UGP^−^ cells. Accordingly, detectable UGP-mediated UDP-Gal production may only occur after a massive increase in the level of Gal1P is attained. Other types of mechanisms are possible and may occur in addition to the mechanism described above. If it is assumed that the very low UGP activity is sufficient to allow some UDP-Glc/UDP-Gal production, then one could speculate that as yet unidentified posttranslational modifications (unaffected by protein synthesis inhibition—see Figure 7B) of enzymes involved in UDP-Glc metabolism may affect cellular levels of UDP-Glc under the different culture conditions used. By way of example, UDP-Glc*N*Ac levels are raised in UGP^−^ cells cultured in the absence of galactose (Figure 6E,F) either because there is little utilization of UDP-Glc*N*Ac (reduced glycoconjugate biosynthesis) or, the reduced UGP activity causes a “back-up” of Glc1P and glucose-6-P which is then diverted through the hexosamine synthesis pathway (see Figure 1) to produce UDP-Glc*N*Ac. Increased UDP-Glc*N*Ac levels can lead to changes in protein stability and localization through posttranslational protein O-Glc*N*Ac modification [32]. Proteins implicated in either the use or production of UDP-Glc could be modified in this manner such that steady state UDP-Glc levels are low. By an as yet unknown mechanism (e.g., raised Gal1P could potentially inhibit an enzyme of the hexosamine biosynthesis pathway), cultivating the cells with galactose could cause UDP-Glc*N*Ac levels to return to normal, reducing protein O-Glc*N*Ac modification and allowing UDP-glucose steady state levels to rise. For example, hyaluronan (HA) synthesis requires both UDP-GlcA and UDP-Glc*N*Ac, and HA synthetase activity is increased by O-Glc*N*Ac modification [33]. Therefore, it is possible that in UGP^−^ cells, cultivated in the absence of galactose, UPD-Glc is preferentially channeled away to HA production via the intermediate of UDP-GlcA.

### 3.3. Defective Glycoconjugate Biosynthesis and Reduced Cellular UDP-Glc/UDP-Gal

We show that treatment of UGP^−^ cells with galactose partially corrects the glycosylation-deficient phenotype. However, we cannot assume that normalization of the glycosylation phenotype of the UGP^−^ cells is uniquely due to restoration of UDP-Glc/Gal levels upon culture of the cells in the presence of galactose. It is also possible that treatment of UGP^−^ cells with galactose may, in addition to raising cellular UDP-Glc/Gal, enable the machineries involved in the different types of glycoconjugate biosynthesis to function more efficiently. Nevertheless, when UGP^−^ cells are treated with galactose, the onset of DLO glucosylation and the rise in cellular UDP-Glc are coincident, and furthermore, when galactose is withdrawn from the culture medium the decline in DLO glucosylation is coincident with the fall in cellular UDP-Glc (Figure 7D,E). Therefore, it is tempting to speculate that, at least, the defective DLO glucosylation in the UGP^−^ cells is solely due to reduced cellular UDP-Glc.

### 3.4. Impaired Protein N-Glycosylation in UGP^−^ Cells

Impaired DLO glucosylation, similar to that seen here with UGP^−^ cells (Figure 5A), causes protein hypoglycosylation in *ALG6*-CDG patients [16]. These patients present with mutations in the *ALG6* gene, which encodes the glucosyltransferase that adds the first glucose residue onto the Man_9_Glc*N*Ac_2_-PP-dolichol *N*-glycosylation precursor. In addition to their failure to glucosylate DLO, UGP^−^ cells do not generate glucosylated *N*-glycans during either pulse radiolabeling experiments or in experiments where these cells were radiolabeled to equilibrium over a 18 h period (Durrant, C. and Moore, S. unpublished data). Under the former conditions, glycoprotein glucosylation reflects both the transfer of Glc_3_Man_9_Glc*N*Ac_2_ from mature DLO onto protein in the ER, and the protein monoglucosylation effectuated by UDP-Glc:glycoprotein glucosyltransferase (UGGT), during glycoprotein folding and quality control [34]. However, as the removal of the three glucose residues from *N*-linked Glc_3_Man_9_Glc*N*Ac_2_ is rapid, the bulk of the Glc_1_Man_9_Glc*N*Ac_2_ observed under the latter experimental (equilibrium) conditions arises from the action of UGGT. This is reinforced by results obtained with either cells from ALG6-CDG patients [16], or ALG6-deficient CHO cells [35,36]. In these cases, although DLO glucosylation is severely perturbed, differences in *N*-glycan monoglucosylation are not observed during longer (greater than 20 min) radiolabeling incubations. Therefore, by contrast to ALG6-deficient cells, both DLO glucosylation and UGGT-mediated protein monoglucosylation are reduced in UGP^−^ cells. Both of these perturbations could potentially contribute to the increased expression of ER stress proteins observed in both the UGP-deficient Don^Q^ and UGP^−^ cell lines [4].

We noted a striking hypoglycosylation of PrP^C^ in Qc cells and demonstrated that the predominant PrP^C^ species appeared to be either unglycosylated or possess a single *N*-glycan. Although protein *N*-glycosylation by oligosaccharytransferase (OST) is thought to be more efficient when the DLO donor is fully glucosylated [37], additional factors such as the folding environment of the ER may contribute to the appearance of PrP^C^ forms that are missing one or two *N*-glycans [38,39,40]. In fact, under certain types of ER stress, a preemptive quality control system has been shown reduce the efficiency with which PrP^C^ is translocated into the ER [39,40]. This process could concieveably lead to the presence of unglycosylated PrP^C^ in the cytosol of UGP^−^ cells. To summarise, further work will be required in order to examine the reasons underlying the appearance of hypoglycosylated PrP^C^ forms in these cells.

### 3.5. Altered Mitochondrial Function in Don^Q^, UGP^−^ and UGP^+^ Cells

Transfection of Don^Q^ cells with UGP leads to complete restoration of cellular UDP-Glc levels but impaired mitochondrial function persists, indicating that the mitochondrial defect is independent of low UDP-Glc levels (Figure 2B). As the hamster Don^Q^ cells were transfected with wild-type bovine UGP, the mitochondrial defect could be explained by postulating a function of UGP that is independent of its UDP-Glc synthesizing ability, and that this second function requires the endogenous hamster enzyme. However, this proposition seems unlikely, as bovine *UGP2* encodes a protein (Uniprot: Q01730) that is 98% identical to its hamster ortholog. It is more likely that the Don^Q^ clone possesses a mutation in a gene implicated in mitochondrial function as well as the G116D mutation in *UGP2*. ATP levels are normal in Don^Q^ cells [41], but this is not unusual as cells possessing only 20% mitochondrial function have been shown to have similar ATP levels and growth rates as control cells [42]. In fact, such cells appear to compensate for their poor oxidative phosphorylation capacities by boosting their ability to generate ATP anaerobically [42].

### 3.6. Altered Cell Growth and Glycosylation Deficits

As well as protein *N*-glycosylation, we show that protein O-glycosylation and glycosphingolipid biosynthesis are severely perturbed in UGP^−^ cells. Despite striking glycosylation deficiencies, the UGP^−^ cells divide slightly faster than UGP^+^ cells. However, it was noted that the latter cell line grows to a higher density than UGP^−^ cells. Cultivation in the presence of galactose also allowed UGP^−^ cells to grow to similar densities to those achieved by UGP^+^ cells and caused changes in the appearance of the UGP^−^ cell monolayer. These differences in cell form were only noted after the cells reached confluence. Glycosphingolipids [43,44] and cell surface glycoproteins [45] modulate cell/cell and cell/extracellular matrix interactions [46]. Taking into account the changes in cellular glycosylation brought about by the galactose treatment, it is tempting to speculate that the more efficient glycosylation of cell surface glycosphingolipids and glycoproteins may normalize these interactions in UGP-deficient cells. After confluence, UGP^−^ cells have reduced viability compared to UGP^+^ cells and cultivating UGP^−^ cells in the presence of galactose does not rescue them from this fate. As mentioned above, we noted that UDP-Glc/UDP-Gal levels fall 2–3-fold during the growth cycle of both UGP^+^ and galactose-treated UGP^−^ cells, and indeed after 8 days in culture medium containing galactose, the UDP-Glc/UDP-Gal levels in UGP^−^ cells were at the limit of detection under our assay procedures (Willemetz, A. and Moore, S. unpublished data).

To conclude, a UGP^−^ cell line displaying reduced levels of UDP-Glc and UDP-Gal has been characterized with respect to its capacity to metabolize galactose and synthesize glycoconjugates. When cultivated in normal growth medium these cells reveal an impaired ability to metabolize a short pulse of [6-^3^H]galactose. Additionally, protein *N*- and *O*-glycosylation as well as glycosphingolipid biosynthesis are severely restricted. Nevertheless, after prolonged treatment with galactose, we show that, by an as yet uncharacterized mechanism, these cells can metabolize galactose, generate UDP-Glc/UDP-Gal, and synthesize glycoconjugates more efficiently than their untreated counterparts. The study of normal and pathological glycoconjugate biosynthesis has been aided by the development of many yeast [47] and mammalian [48] cell experimental models in which genes have been invalidated by either homologous recombination or mutagenesis. Here, we describe an addition to the list of such mammalian glycosylation mutants, and the capacity of the UGP^−^ cell line to change its glycosylation status as a function of extracellular galactose makes it a useful model with which to study different aspects of galactose metabolism and glycoconjugate biosynthesis. Recently, after the completion of this work, *UGP2* mutations in the human population were reported to underlie some forms of developmental epileptic encephalopathy (DEE) [8]. Our data suggest that a careful study of galactose metabolism in cells from these patients will be required in order to decipher cellular pathology mechanisms and may suggest potential galactose-related treatment strategies for these patients.

## 4. Materials and Methods

All reagents were purchased from SIGMA-Aldrich SARL, Saint Quentin Fallavier, FR, unless otherwise stated.

### 4.1. Generation, Culture and Radiolabeling of Cells

The UGP-deficient cell line (Don^Q^) was generated by cultivating mutagenized Chinese hamster lung fibroblasts (Don) with *Clostridium difficile* toxin B [7]. Don^Q^ cells stably transfected with either the pCDNA3 vector containing the wild-type bovine *UGP2* gene (G3 cells), or empty vector (Qc cells) have also been described [6]. All cells were cultivated in RPMI 1640 (Invitrogen, Cergy Pontoise, FR) containing 10% fetal calf serum (Invitrogen), 1% penicillin-streptomycin and 0.4 mg/mL G-418 sulphate. For short radiolabeling periods, cells were incubated with either 200 µCi D-[2-^3^H]mannose (20 Ci/mmol, Perkin Elmer, Courtaboeuf, FR) or 100 µCi [6-^3^H]galactose (40 Ci/mmol, BioTrend, Köln, DE) for 20 min in glucose-free RPMI 1640 (Invitrogen) supplemented with 5% dialyzed FCS and 0.5 mM glucose. For equilibrium radiolabeling, cells were incubated with 200 µCi D-[2-^3^H]mannose for 18 hours in 5 mL glucose-free RPMI 1640 supplemented with 5% dialyzed FCS, 5 mM fucose, 0.5 mM glucose and 1 mM galactose. As detailed in the figure legends, media with different levels of added glucose and galactose are used. These media were made by adding glucose and galactose to glucose-free RPMI 1640 (Invitrogen) containing 10% undialyzed FCS. The glucose concentration in glucose-free RPMI 1640 containing 10% undialyzed FCS alone was determined to be ~0.5 mM.

### 4.2. Measurement of Mitochondrial Complex II, III and IV Activities

All spectrophotometric methods have been described [49] and were performed on freeze-thawed cells (one cycle) in 1 mL appropriate buffer using a Varian Cary 50 Bio spectrophotometer. Cytochrome c oxidase (complex IV) was measured at 550–540 nm by following the oxidation of reduced cytochrome c in 10 mM phosphate buffer, pH 6.5, in the presence of 4 µM lauryl-maltoside. Complex III was assayed at 550–540 nm in 10 mM phosphate buffer, pH 7.8, containing 2 mM EDTA and 0.4 µM lauryl-maltoside by following reduction of cytochrome c upon addition of 50 µM decyl-ubiquinol. Complex II activity was obtained by measuring the rate of dichlorophenol indophenol (100 µM) reduction triggered by 5 mM succinate in the same buffer as for complex III, but at 600–750 nm and in the presence of 40 µM of decyl-ubiquinone.

### 4.3. Assay of Nucleotide Sugars

Nucleotide sugars were determined after solid-phase (SPE) extraction and HPLC as previously described [18]. Cells were seeded (1.0–2.5 × 10^5^ cells per 25 cm^2^ flask) and grown under different conditions for 3–5 days prior to being extracted in ice-cold 75% ethanol. After centrifugation, supernatants were dried and loaded onto Supelclean^TM^ ENVI^TM^-Carb SPE (3 mL) tubes (SIGMA-Aldrich SARL, Saint Quentin Fallavier, FR) in 5 mL 10 mM ammonium bicarbonate. The tubes were washed with the following solutions in the indicated order: 2 mL H_2_O, 2 mL 25% Ac*N* in H_2_O, and 2 mL 50 mM triethylammonium acetate (TEAA), pH 7.0, before eluting sugar nucleotides with 4 mL 25% Ac*N* in 50 mM TEAA, pH 7.0. The sugar nucleotides were analyzed by HPLC (Waters 600 solvent delivery system) on a reversed phase column (LiChrosorb RP-18 5 µm (250 mm × 4.6 mm, SIGMA-Aldrich SARL) equilibrated in 20 mM TEAA, pH 6.0. Sugar nucleotides were separated by eluting the column in the same buffer at 0.5 mL/min, and then detected by spectrophotometry (Waters 2487 Dual Wavelength Detector, Waters Corporation, Milford MA, USA) at 254 nm. Radioactive sugar nucleotides were detected with a Packard 150 TR flow-scintillation analyzer (Perkin Elmer), using Optima-Flo AP (Perkin Elmer) scintillation cocktail at a flow rate of 1 mL/min.

### 4.4. Assay of Sugar Phosphates

Material not retained by Supelclean^TM^ ENVI^TM^-Carb SPE cartridges was dried and loaded onto DEAE cellulose columns. After washing the columns, sugar phosphates were eluted with 500 mM NH_4_HCO_3_. After alkaline phosphatase treatment and desalting on AG-1/AG-50 columns, radioactive material was resolved on cellulose-coated plastic TLC plates developed in pyridine/ethyl acetate/H_2_O/acetic acid, 5:5:3:1 (solvent system A) and detected by fluorography after spraying the dried TLC plates with En^3^hance^®^ (Perkin Elmer).

### 4.5. Analysis of Glycosphingolipids

Lipids were extracted by the tetrahydrofuran/phosphate buffer method [50], using 4 × 75 cm^2^ dishes, for each sample. The aqueous phase obtained, containing gangliosides, was evaporated to dryness, resuspended, and exhaustively dialyzed against distilled water before drying. The gangliosides, resuspended in chloroform/methanol, 2/1, by vol, were analyzed using HPTLC (Silica gel 60 TLC) plates developed in chloroform/methanol/0.2% acqueous CaCl_2_ 55/42/11, by vol (solvent system B). Visualization was performed with *p*-dimethylaminobenzaldehyde spray reagent. The organic phase obtained, containing neutral glycosphingolipids, was evaporated to dryness. Lipids, after mild alkaline hydrolysis, were dialyzed, collected and finally dried. The neutral glycosphingolipids were resuspended in chloroform/methanol, 4/1, by vol, and analyzed using HPTLC (Silica gel 60 TLC) plates developed in chloroform/methanol/formic acid/water, 65/25/8.9/1.1, by vol (solvent system C). Visualization was performed with a 10% CuSO_4_/8% phosphoric acid aqueous spraying solution.

### 4.6. Preparation and Purification of Dolichol-Linked Oligosaccharides (DLO)

Cells were extracted by a modification of the method of Folch [51,52]. Briefly, cells were extracted with chloroform/methanol/125 mM Tri/HCl (pH 7.2) containing 4 mM MgCl_2_, 3:2:1, and after vigorous shaking, the upper methanolic and the lower chloroformic phases were removed and kept, and the interphase proteins were extracted with 2 × 2 mL CHCl_3_/MeOH/H_2_O (10:10:3). The chloroform and 10:10:3 phases were dried and hydrolyzed with 0.02 N HCl in order to release oligosaccharides from DLO. After desalting on AG-1/AG-50 columns, oligosaccharidic components were resolved by TLC on silica-coated plastic sheets in n-propyl-alcohol/acetic acid/H_2_O, 3:3:2 (solvent system D) for 36 h and detected by fluorography after spraying the dried TLC plates with En^3^hance^®^.

### 4.7. Polymannose-Type N-Glycan Analysis

Protein pellets were treated with pronase (2 mg/mL in 50 mM ammonium bicarbonate (pH 8.0) buffer for 24 h at 37 °C. Reaction mixtures were loaded onto AG1-X2/ AG50-X2 (Bio-Rad Laboratories, Marnes-La-Coquette, FR) columns which were washed with H_2_O before eluting glycopeptides with 2M pyridine acetate (pH, 4.0). Glycans were released from aliquots of glycopeptides by treating with 1.5 mU of endo H in 20 µL sodium citrate buffer, pH 5.5, for 15 h at 37 °C [53]. The reaction mixtures were desalted on combined AG1-X2/AG50-X2 columns, and neutral oligosaccharides were resolved by TLC on silica-coated plastic sheets developed in solvent system D and detected as described above.

### 4.8. Lectin Affinity Chromatography

Aliquots of glycopeptides were subjected to ConA-Sepharose (GE Healthcare, Vélizy Villacoublay, FR) chromatography as described previously [54,55]. The presence of terminal non-reducing galactose residues on *N*-glycans was evaluated by subjecting the [2-^3^H]mannose labeled glycopeptides to *Ricinus communis* agglutinin I (RCA-I, E.Y Laboratories, San Mateo, CA, USA) lectin chromatography [56]. Prior to analysis, glycopeptides were treated with 100 µL 50 mM H_2_SO_4_ for 1 h at 80 °C. After cooling and neutralization with barium acetate, glycopeptides were applied to the column in 50 mM Tris/HCl (pH 7.8) buffer containing 100 mM NaCl, 1 mM CaCl_2_, 1 mM MgCl_2_ and 0.02% (*w*/*v*) sodium azide (LB). The column was subsequently eluted with 10 mL LB (fraction I), 10 mL LB containing 0.1 mM lactose (fraction II), 10 mL LB containing 1.0 mM lactose (fraction III), and finally 10 mL LB containing 10 mM lactose (fraction IV).

### 4.9. Lectinoblot and Western Blot Analysis

Cells were scraped from tissue culture flasks with PBS and collected by centrifugation (10 min, 1200× *g*/min). Cell pellets were resuspended in lysis buffer (1% Triton X-100, 25 mM Tris-HCl pH 7.4, 1 mM PMSF, 1% protease inhibitor cocktail (SIGMA-Aldrich SARL), 1 mM Na_3_VO_4_, 1 mM NaF) for 1 h at 4 °C. The cell lysate was centrifuged (100,000× *g*/min, 1 h, 4 °C) and the supernatant was recovered. Protein concentrations were determined by the bicinchoninic acid method (BCA, [57] as recommended by the manufacturer (Pierce). For detection of cell surface PrP^C^, cells were washed with PBS, and then labeled for 30 min at 4 °C with EZ-link^TM^-sulfo NHS-LC-biotin (Thermo Fisher Scientific, Cergy Pontoise, FR) at a concentration of 0.5 mg/mL in PBS. Biotinylated cells were washed then lysed as described above. Immunoprecipitation was performed using protein A-Sepharose beads coupled to PrP^C^ (a.a. 79–92) specific monoclonal antibody, SAF32 (SPIBIO, Montigny le Bretonneux, FR). Beads were washed and PrP^C^ was eluted with denaturating SDS-sample buffer. Soluble PrP^C^ in the cell culture supernatant was also analyzed. For this purpose, cell cultures were incubated overnight in serum free medium. Cell debris were eliminated prior to ethanol precipitation of proteins from culture supernatant. Extracted or immunoprecipitated or ethanol precipitated proteins were submitted to SDS-polyacrylamide (10%) gel electrophoresis and then electrotransferred onto nitrocellulose membranes. Lectinoblots were performed as previously described [58] using biotinylated RCA-I/avidin-peroxidase or Jacalin-peroxidase (EY Laboratories, CA, USA). Western blot analysis of the Tn antigen was performed as described [59], and the MAb 83D4 [60] was kindly provided by Eduardo Osinaga (Depto. de Bioquímica, Facultad de Medicina, Montevideo, Uruguay). Bound primary antibodies were detected by chemiluminescence (ECL, GE Healthcare) using horseradish peroxidase (HRP)-conjugated secondary antibodies against rabbit or mouse immunoglobulins. For Western blot analysis of cellular PrP^C^ nitrocellulose membranes were blocked with 1% gelatin in PBS 0.1% Tween 20, incubated with 10 μg/mL SAF32 antibody, and revealed using secondary antibodies coupled to HRP. Western blot analysis of biotinylated PrP^C^ was performed as above, except that biotinylated PrP^C^ was detected by streptavidin-HRP.

### 4.10. Cell Surface Immunofluorescence Studies

UGP^+^ and UGP^−^ cells grown on glass coverslips in 24 well micro plates were incubated in either the presence or absence of 10 mM galactose. Cell surface immunofluorescence was carried out on living cells, which were incubated at room temperature with 10 μg/mL of SAF32 in PBS, 2% FCS and 0.1% sodium azide to avoid internalization of cell surface proteins. After 3 washes, fluorescent antibodies were added for 1 h. Cells were fixed with 3.7% formaldehyde, and then mounted in fluoromount (CliniSciences SAS, Montrouge, FR) before examination on an Axiophot microscope (Zeiss) equipped with a UV lamp and appropriate emission filters for epifluorescence and with a camera (Nikon) and video system (Packard Bell).

### 4.11. Generation of HsUGP(G116D) and Production of Recombinant Wild-Type and Mutant HsUGPp

A previously generated expression plasmid for N-terminally StrepII-tagged HsUGP was used to recombinantly produce the full-length wild-type enzyme as previously described [24]. Using this plasmid as a template, the G116D mutation was introduced employing the same strategy as described earlier [24]. The integrity of the resulting plasmid and the presence of the desired mutation was confirmed by sequencing of the plasmid (GATC Biotech, Konstanz, DE). Recombinant expression in *Escherichia coli*, affinity chromatographic purification, buffer exchange and storage of both the wild-type and mutant enzyme were likewise carried out as described before, and the purity and integrity of the proteins were confirmed by SDS-PAGE followed by Coomassie protein staining or Western blot analysis [24].

### 4.12. In Vitro Activity Assay and Determination of Kinetic Parameters

Analysis of enzymatic activity and determination of kinetic parameters in the forward direction of the UGP reaction (production of UDP-Glc and PP_i_), including curve fitting and evaluation using GraphPad Prism 4 software, were performed as described before [24], with the following adaptations: UTP concentration was varied between 0.01 and 0.8 mM at a constant concentration of 2 mM of Glc1P or Gal1P; and concentrations of Glc1P, or Gal1P, were varied between 0.01 and 1 mM, or 0.01 and 2 mM, respectively, at a constant concentration of 1 mM UTP.

## Figures and Tables

**Figure 1 ijms-21-02028-f001:**
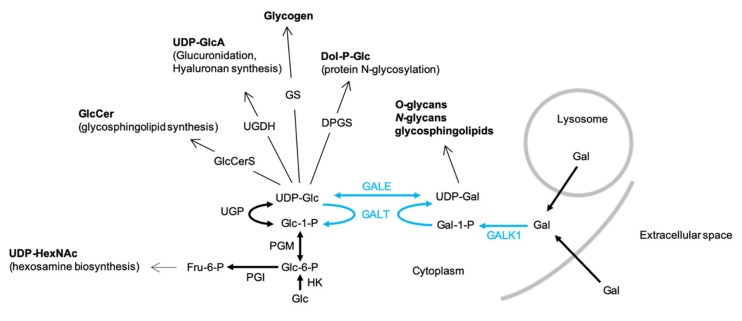
The biosynthesis and physiological roles of UDP-glucose (UDP-Glc) and UDP-Gal-UDP-Glc and UDP-Gal are utilized in many biosynthetic pathways: GlcCerS, glucosylceramide synthase; UGDH, UDP-glucose dehydrogenase; GS, glycogen synthase; DPGS, dolichol-P-glucose synthase. UDP-Glc is generated from either UDP-Gal by UDP-galactose 4-epimerase (GALE) or from glucose-1-phosphate (Glc1P) and UTP by UDP-Glc pyrophosphorylase (UGP). The synthesis of UDP-Gal via the Leloir pathway (blue arrows) itself requires UDP-Glc. After transport into the cell, galactose (Gal) is phosphorylated by galactokinase (GALK1) to yield galactose-1-phosphate (Gal1P), which in turn is converted into UDP-Gal by galactose-1-phosphate uridyltransferase (GALT), whose uridyl donor is UDP-Glc. UGP deficiency could potentially provoke increased fluxing of Glc-6-P through the hexosamine biosynthesis pathway to yield increases in cellular UDP-GlcNAc/UDP-GalNAc (UDP-HexNAc). The bold arrows represent single reaction steps whereas normal arrows indicate pathways involving several steps.

**Figure 2 ijms-21-02028-f002:**
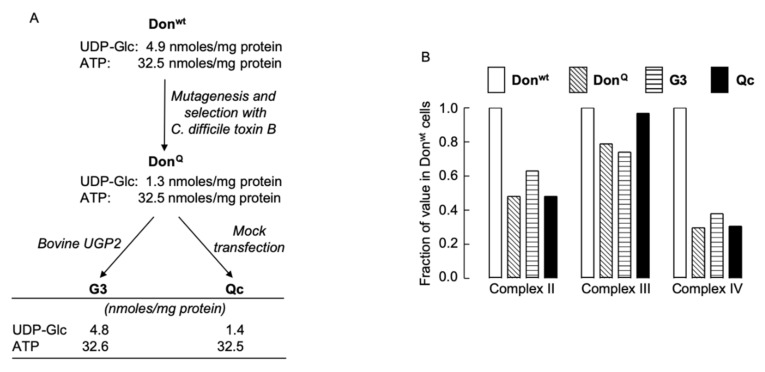
Derivation of UGP-deficient cell lines and examination of their mitochondrial function. (**A**). Chinese hamster lung fibroblasts (Don^wt^) were mutagenized with ethylmethanesulphonate, and clones resistant to *Clostridium difficile* (*C. difficile*) toxin B were selected [7]. One clone (Don^Q^) was found to harbor a point mutation (G116D) in *UGP2* [5]. Don^Q^ cells were transfected with wild-type bovine *UGP2* (G3 cells) or mock transfected (Qc cells). (**B**). The four cell lines described above were cultivated in complete growth medium for five days and enzyme activities associated with mitochondrial complexes II, III, and IV were measured as described in Materials and Methods. The activities found in Don^wt^ cell extracts were set at 1.

**Figure 3 ijms-21-02028-f003:**
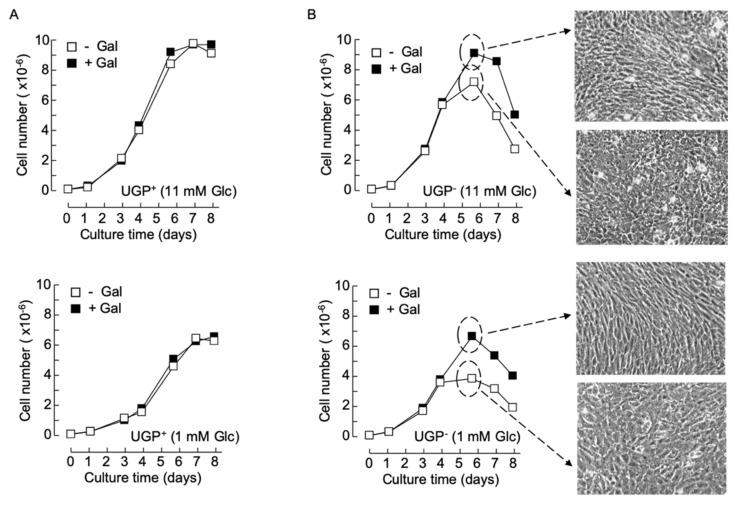
Cell growth and morphology changes in galactose-cultivated UGP^−^ cells – UGP^+^ (**A**) and UGP^−^ (**B**) cells were cultivated in media containing either 11 mM Glc (− Gal) or 11 mM Glc + 10 mM Gal (+ Gal) (**A**, upper panel and **B**, upper panel), or 1 mM Glc (− Gal) or 1 mM Glc + 5 mM Gal (+ Gal) (**A**, lower panel and **B**, lower panel) for 8 days. At the indicated times, cells were released from tissue culture flasks with trypsin and counted. Each growth curve is from a single experiment. On day 5 (dotted circles and arrows), the appearance of the UGP^−^ cell monolayers was recorded using phase contrast microscopy (magnification × 10).

**Figure 4 ijms-21-02028-f004:**
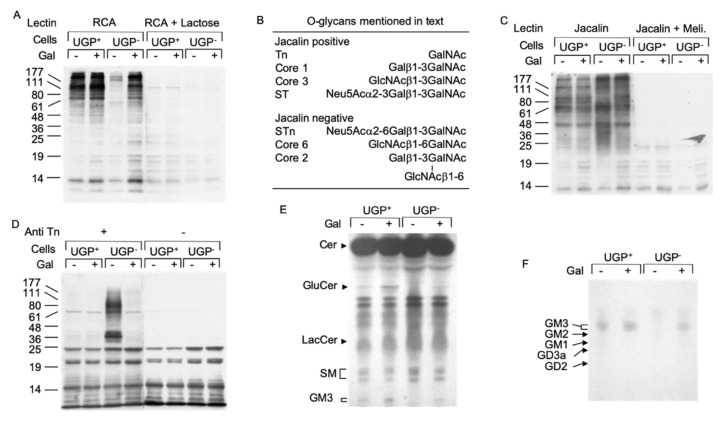
Deficits in protein *O*-glycosylation and glycosphingolipid biosynthesis in UGP^−^ cells. (**A**). UGP^+^ and UGP^−^ cells were incubated in media containing either 1 mM Glc or 1 mM Glc + 5 mM Gal for 24 h. Cell extracts were analyzed by SDS PAGE and after transfer of proteins onto nitrocellulose membranes, glycoproteins bearing terminal non-reducing galactose residues were probed using biotinylated *Ricin communis* agglutinin I (RCA). To control for the specificity of lectin binding, a duplicate membrane was incubated with the lectin in the presence of 500 mM lactose. Bound lectin was detected using avidin peroxidase/ECL reagent. The migration positions of standard molecular weight (numbers in Kd) markers are indicated to the left of the blot. (**B**). As reported [12], Jacalin lectin recognizes some but not all *O*-glycans. (**C**). The same cell extracts as in (**A**) were analyzed as described above and nitrocellulose membranes were incubated with Horseradish peroxidase conjugated Jacalin lectin in either the absence or presence of 125 mM melibiose (Meli). (**D**). These extracts were analyzed as described above and membranes were probed with an antibody directed against the Tn antigen (Anti Tn). (**E**). UGP^+^ and UGP^−^ cells were cultivated for 5 days in media to which either 1 mM Glc or 1 mM Glc + 5 mM Gal was added. The cells were harvested and glycosphingolipds were extracted as described in Material and Methods. Neutral species were resolved by TLC using solvent system B. (**F**). Charged glycosphingolipids were resolved by TLC using solvent system C. The abbreviations used are: Cer, ceramide; GlcCer, Glcβ-ceramide; LacCer, Galβ1,4Glcβ-ceramide; SM, sphingomyelin; GM3, Neu5Ac(α2,3)Galβ1,4Glcβ-ceramide; GM2, GalNAcβ(1–4)[Neu5Ac(α2,3)]Galβ1,4Glcβ-ceramide; GM1, Galβ(1–3)GalNAcβ(1–4)[Neu5Ac(α2,3)]Galβ1,4Glcβ-ceramide; GD2, GalNAcα(1–4)[Neu5Ac(α2,8)Neu5Ac(α2,3)]Galβ1,4Glcβ-ceramide; GD3, Neu5Ac(α2,8)Neu5Ac(α2,3)]Galβ1,4Glcβ-ceramide.

**Figure 5 ijms-21-02028-f005:**
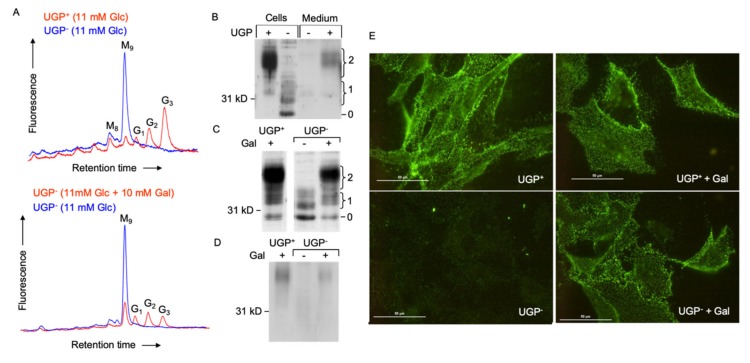
Dolichol-linked oligosaccharide biosynthesis and *N*-glycosylation of the normal form of the prion protein (PrP^C^) in UGP^−^ and UGP^+^ cells. (**A**). Upper panel: UGP^+^ and UGP^−^ cells cultivated in RPMI 1640 growth medium. Lower panel: UGP^−^ cells were incubated in glucose-free RPMI 1640 media containing the indicated levels of glucose and galactose for 24 h. Cells were harvested, and dolichol-linked oligosaccharides were extracted with organic solvents. After acid hydrolysis, liberated oligosaccharides were derivatized with 2-aminopyridine and analyzed by HPLC as described in Materials and Methods. The retention times of standard oligosaccharides are indicated: G_3_, Glc_3_Man_9_Glc*N*Ac; G_2_, Glc_2_Man_9_Glc*N*Ac; G_1_, Glc_1_Man_9_Glc*N*Ac; M_9_, Man_9_Glc*N*Ac; M_8_, Man_8_Glc*N*Ac. (**B**). UGP^+^ and UGP^−^ cells were grown for 5 days in normal growth medium prior to submitting cell extracts and media to SDS-PAGE. Blotted proteins were probed with an antibody that recognizes PrP^C^ irrespective of its glycosylation status. The curly brackets group bands known to correspond to PrP^C^ glycoforms possessing 1 or 2 *N*-glycans. The migration position of the non-glycosylated prion protein is indicated (0). (**C**). Cells were cultivated in normal growth medium (Gal-) or normal growth medium supplemented with 10 mM galactose (Gal+), as indicated, for two days prior to harvesting cells and submitting cellular extracts to SDS PAGE as described for panel A. (**D**). After cell surface biotinylation, PrP^C^ was immunoprecipitated and submitted to SDS PAGE. Subsequent to Western blot analysis, biotinylated PrP^C^ was detected using HRP-streptavidin. (**E**). After culturing cells as described above, cell surface PrP^C^ was monitored in living cells by immunofluorescence microscopy. The scale bar represents 50 µm.

**Figure 6 ijms-21-02028-f006:**
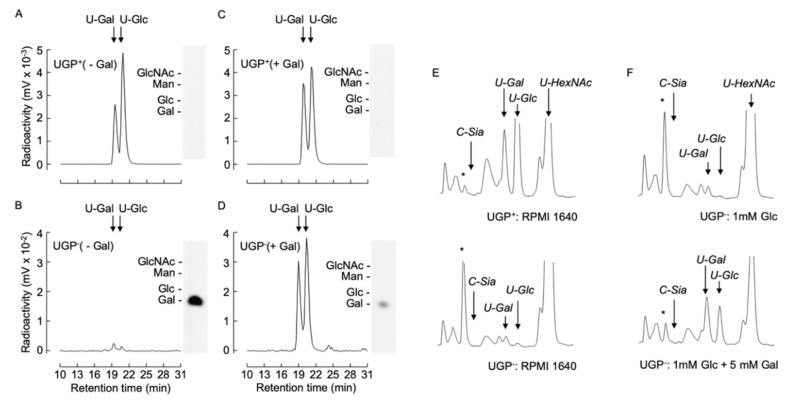
[6-^3^H]galactose metabolism and UDP-Glc/Gal steady state levels in UGP^+^ and UGP^−^ cells. (**A**–**D**). UGP^+^ cells (**A**,**C**) and UGP^−^ cells (**B**,**D**) were incubated for 16 h in glucose-free medium to which either 1 mM Glc (- Gal, **A**,**B**) or 1 mM Glc and 5mM Gal (+ Gal, **C**,**D**) was added. Cells were then metabolically radiolabeled with [6-^3^H]galactose for 30 min as described in Material and Methods. Then 75% ethanol extracts were clarified by centrifugation and soluble material was dried and submitted to SPE solid-phase extraction. The SPE run through material was subjected to DEAE Sephadex chromatography and after elimination of unbound material, negatively charged components were eluted and subjected to alkaline phosphatase digestion prior to resolution by TLC (insets). The standards used are: *N*-acetylglucosamine, Glc*N*Ac; glucose, Glc; galactose, Gal; mannose, Man. Sugar nucleotides eluted from the SPE were dried down and subjected to HPLC as described in Materials and Methods. (**E**). UGP^+^ and UGP^−^ cells were grown in normal growth medium for 4 days and nucleotide sugars were extracted and separated as described above and detected by spectrophotometry at 524 nm. (**F**). UGP^−^ cells were grown as described above prior to incubating cells for 6 h in glucose-free medium to which either 1 mM Glc or 1 mM Glc and 5mM Gal was added. The quantity of material injected was normalized according to the protein content of the cellular extracts. The arrows indicate the elution times for: CMP-Neu5Ac, *C-Sia*; UDP-galactose, *U-Gal*; UDP-glucose, *U-Glc*; UDP-N-acetylglucosamine/UDP-N-acetylgalactosamine, *U-HexNAc*. The asterisks indicate the elution position of an unknown component that is discussed in the text.

**Figure 7 ijms-21-02028-f007:**
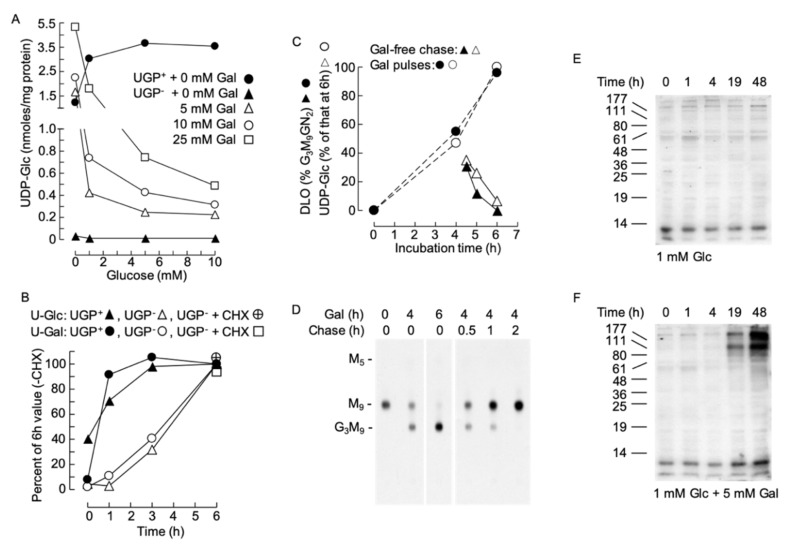
(**A**). UGP^+^ and UGP^−^ cells were grown as described above prior to being treated for 6 h with glucose-free medium to which the indicated additions of glucose and galactose were made. UDP-Glc was measured as described above. (**B**). UGP^+^ and UGP^−^ cells were cultivated for 4 days in normal growth medium prior to incubating the cells, for the indicated times, with glucose-free medium to which 1 mM Glc and 5 mM Gal were added. In one experiment the cells were pretreated for 30 min with 10 µg/ml cycloheximide (CHX). UDP-Glc (*U-Glc*) and UDP-Gal (*U-Gal*) were measured as above. The quantities of these nucleotide sugars have been expressed as a percentage of the quantity found to occur at the 6 h time point in the absence of CHX. All data points represent a single determination. (**C**,**D**). UGP^−^ cells were grown in normal growth medium for 4 days, washed in glucose-free RPMI 1640 and then incubated for the indicated times in this medium containing 1 mM Glc + 5 mM Gal (Gal). Where indicated, the cells were then rinsed with glucose-free RPMI 1640 prior to being incubated for the indicated times in normal growth medium (chase). Subsequent to these incubations, UDP-Glc was measured. The observed UDP-Glc levels have been expressed as a percentage of that occurring at 6 h of Gal treatment. All data points represent a single determination. In a parallel experiment, cells were radiolabeled with [2-^3^H]mannose for 30 min and DLO were extracted and examined by thin-layer chromatography (**D**). The fluorograph shown in D was scanned densitometrically and the percentage of total DLO occurring as Glc_3_Man_9_Glc*N*Ac_2_ was computed. F. UGP^−^ cells were incubated for up to 48 h in the indicated media. After harvesting, the galactosylation of cellular proteins was examined by RCA lectinoblot as described for Figure 4A. This is a representative experiment, and in another experiment, a control membrane, where the blot was incubated with RCA-I in the presence of 500 mM lactose, revealed that detection of bands was sugar dependent (as demonstrated in Figure 4A).

**Figure 8 ijms-21-02028-f008:**
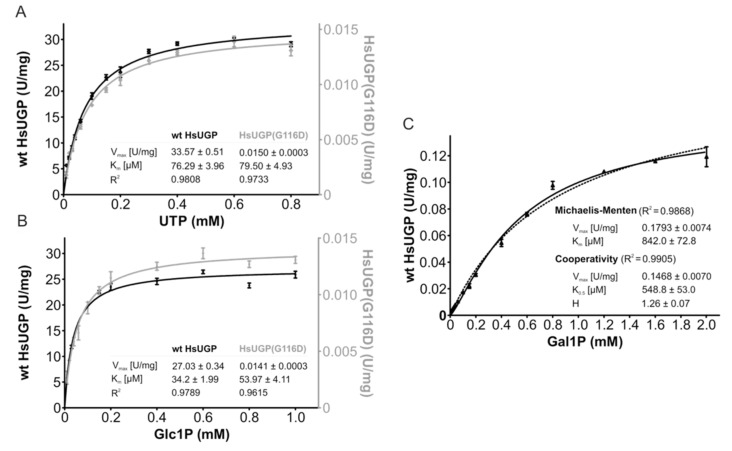
Enzymatic activity and kinetic parameters of recombinant wild-type HsUGP and HsUGP(G116D). (**A**,**B**). In vitro activity of recombinant purified wild-type HsUGP (black) and HsUGP(G116D) (grey) as a function of UTP (**A**) or Glc1P (**B**) concentration. (**C**). In vitro activity of recombinant purified wild-type HsUGP as a function of Gal1P concentration (activity of the G116D variant with this substrate was not detectable). Data points represent the mean of six determinations. Curve fitting according to the Michaelis–Menten model (**A**–**C**) or the cooperativity model (**C**) and determination of kinetic parameters (reported in the insets as the means +/− standard errors of the means) were performed in GraphPad Prism 4.

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
