# Peer review of "Defects in Galactose Metabolism and Glycoconjugate Biosynthesis in a UDP-Glucose Pyrophosphorylase-Deficient Cell Line Are Reversed by Adding Galactose to the Growth Medium"

_ijms, 2020, doi:10.3390/ijms21062028_

Round 1

Reviewer 1 Report

The authors of this manuscript set off to characterize a variant strain of Chinese hamster lung fibroblast line called DonQ and its derivatives using a host of biochemical assays.  The strengths of the manuscript lie on the interesting/intriguing finding that although DonQ cannot survive in galactose-only medium, galactose addition to glucose medium actually improved many of its biochemical deficits.  In addition, the in-depth characterization, which include extensive lectin blots and immunohistochemical studies, yield some interesting results.

The weaknesses of the work stem from the lack of any rationale and focus behind the study.  Are they trying to understand the mechanisms behind the galactose rescue?  If so, their characterization did not turn up any novel mechanistic insights.  Moreover, this reviewer wonders if the use of up to 10mM galactose was physiological relevant.  The authors argued that this work will benefit patients with DEE, but if this is the goal, will a neuronal cell line (instead of lung fibroblasts) be more appropriate?  Also, in the beginning of the manuscript, the authors looked at Complex II and IV activities (Fig 2b).  They concluded that the mitochondrial defects in the mutant cells is not correlated with UDP-Glc level reduction.  This raised the question of whether the (biallelic ?) hypomorphic G116D mutation is the only mutation in this cell line.  Have the authors sequenced the genome of DonQ to confirm that this is the only mutation resulted from the mutagenesis?

In Figure 1, the word galactokinase was mis-spelled and the conventional abbreviation for it should be GALK1 (not GK).

Author Response

REVIEWER 1

“The weaknesses of the work stem from the lack of any rationale and focus behind the study”.

Response

In order to help understand the rationale of the experimental approach, the below information has now been included in a new results subchapter of the revised manuscript (2.1. Rationale for the study).

(Starts line 82) The dolichol linked oligosaccharide (DLO) required for protein N-glycosylation is glucosylated by enzymes that use dolichyl-P-glucose (DPG) as donor. The biosynthesis of DPG in turn requires UDP-Glc and led us to ask whether or not conditions that lead to changes in cellular UDP-Glc levels could affect DLO glucosylation and therefore regulate protein N-glycosylation in normal cells. Accordingly, characterisation of DLO glucosylation and protein N-glycosylation in UGP-deficient CHL CgUGP(G116D) fibroblasts posessing 30 % normal UDP-glucose levels [5] was of interest. In preliminary experiments we could not detect DLO glucosylation in these cells (Durrant, C. and Moore, S., unpublished observations), and while attempting to boost UDP-glucose levels by supplementing the culture medium with mixtures of uridine and various monosaccharides it was noted that supplementation of the culture medium with galactose alone was sufficient to allow complete DLO glucosylation (Durrant, C. and Moore, S., unpublished observations).

This result, coupled with the fact that these cells are unable to survive when galactose is the sole energy source[5], led us to initiate a study of galactose and glycoconjugate metabolism in these cells and provide a global picture of these metabolic pathways in UGP-deficient cells.

 “Are they trying to understand the mechanisms behind the galactose rescue?  If so, their characterization did not turn up any novel mechanistic insights”. 

Response

The mechanism of the galactose reversal phenomenon was not addressed, although we did rule out the intriguing possibility that the UGP2(G116D) variant prefers Gal1P over Glc1P as substrate. Understanding the mechanism will require careful metabolic fluxing experiments coupled with enzyme assays performed in the presence of different metabolite concentrations. This will be a large undertaking and is not within the scope of the present manuscript.

“Moreover, this reviewer wonders if the use of up to 10mM galactose was physiological relevant”.

Response

5 -10 mM galactose may not be physiologically relevant, but it has been shown that serum galactose reaches about 2 mM in phosphoglucomutase 1 deficient CDG patients (PGM1-CDG) undergoing 1 g galactose/Kg/day therapy (Radenkovic S, et al. Am J Hum Genet 2019. PMID 30982613). In fact, as shown in the Supplementary Table of our manuscript, a 1 mM galactose / 0.5 mM glucose regimen can reverse N-glycosylation deficits (see also Materials and Methods for the steady-state radiolabelling protocol for this experiment).

“The authors argued that this work will benefit patients with DEE, but if this is the goal, will a neuronal cell line (instead of lung fibroblasts) be more appropriate”?

Response

This was not the goal of our work: DEE patients with UGP2 deficiency were first reported a few months ago (December 2019) whereas our study was initiated many years ago. Clearly, it will now be important to look at galactose metabolism in neuronal cell models of UGP2 deficiency.

“Also, in the beginning of the manuscript, the authors looked at Complex II and IV activities (Fig 2b).  They concluded that the mitochondrial defects in the mutant cells is not correlated with UDP-Glc level reduction.  This raised the question of whether the (biallelic?) hypomorphic G116D mutation is the only mutation in this cell line.  Have the authors sequenced the genome of DonQ to confirm that this is the only mutation resulted from the mutagenesis”?

Response

We have not sequenced the entire genome of DonQ. Our experiments with complex II and IV suggest that it is unlikely that the UGP2(G116D) mutation is the only mutation in DonQ cells. In fact, this is one of the draw backs with working with cell lines that have been derived from a population of mutagenized cells. Nevertheless, this problem is common to many similar cell lines (eg CHO-derived lectin resistant cell lines, which have nevertheless proved valuable tools for glycobiology research (Stanley, P. (1989) Mol Cell Biol. 9: 377–383; Ermonval M, (1997) et al. J Cell Sci 110: 323–336).

“In Figure 1, the word galactokinase was mis-spelled and the conventional abbreviation for it should be GALK1 (not GK)”.

Response: this has been corrected in Figure 1 and the figure legend of the revised manuscript.

Reviewer 2 Report

An interesting article describing several experiments with CHLF cells harbouring a mutation resulting in a reduction of UDP-glucose pyrophosphorylase activity to some 5% of normal.

Culture of the cells  in galactose restricted medium supplemented only with glucose - resulted in reduced levels of UDP-glucose, UDP-galactose as well as reduced glycoprotein and glycolipid synthesis which need UDP-glucose or UDPglactose as substrates and these reductions were reversed by the addition of galactose in the medium. The metabolic pathway for this "resque" by galactose still remains unknown and it would have been interesting if the authors could have speculated a bit more around this.

The article is clearly written although some of the figures are very small. The F is missing on panel 7F but that is not a problem.

Author Response

REVIEWER 2

“The metabolic pathway for this "resque" by galactose still remains unknown and it would have been interesting if the authors could have speculated a bit more around this”.

Response

We have now added a paragraph to the discussion of the revised manuscript in which we speculate about the mechanism of the galactose “rescue”. (Starts line 493) The delay in the onset of the galactose rescue phenomenon maybe related to the accumulation of Gal1P in galactose-treated UGP- cells. Accordingly, significant UGP-mediated UDP-Gal production may only occur after a massive increase in the level of Gal1P is attained. Other types of mechanism are possible and may occur in addition to the mechanism described above. If it is assumed that the very low UGP activity is sufficient to allow some UDP-Glc/UDP-Gal production then one could speculate that as yet unidentified posttranslational modifications (unaffected by protein synthesis inhibition – see Figure 7B) of enzymes involved in UDP-Glc metabolism may affect cellular levels of UDP-Glc under the different culture conditions used. By way of example, UDP-GlcNAc levels are raised in UGP- cells cultured in the absence of galactose (Figure 6E and F) either because there is little utilisation of UDP-GlcNAc (reduced glycoconjugate biosynthesis) or, the reduced UGP activity causes a “back-up” of Glc1P and glucose-6-P which is then diverted through the hexosamine synthesis pathway (see Figure 1) to produce UDP-GlcNAc. Increased UDP-GlcNAc levels can lead to changes in protein stability and localisation through posttranslational protein O-GlcNAc modification [32]. Proteins implicated in either the use or production of UDP-Glc could be modified in this manner such that steady state UDP-Glc levels are low. By an as yet unknown mechanism (eg. Raised Gal1P could potentially inhibit an enzyme of the hexosamine biosynthesis pathway), cultivating the cells with galactose could cause UDP-GlcNAc levels to return to normal, reducing protein O-GlcNAc modification and allowing UDP-glucose steady state levels to rise. For example, hyaluronan (HA) synthesis requires both UDP-GlcA and UDP-GlcNAc, and HA synthetase activity is increased by O-GlcNAc modification [33]. Therefore, it is possible that in UGP- cells, cultivated in the absence of galactose, UPD-Glc is preferentially channelled away to HA production via the intermediate of UDP-GlcA.

“The article is clearly written although some of the figures are very small. The F is missing on panel 7F but that is not a problem”.

Response:

I understand that Figure 8 was difficult to evaluate (also in Editor’s comments) because error bars and the statistical analysis are not evident, so this figure has been enlarged. The F has now been added to Panel 7F.